Manuscript prepared for Biogeosciences
with version 2014/09/16 7.15 Copernicus papers of the LaTeX class copernicus.cls.
Date: 13 January 2017

# Flying the satellite into your model: on the role of observation operators in constraining models of the earth system and the carbon cycle

Thomas Kaminski[1] and Pierre-Philippe Mathieu[2]

[1]The Inversion Lab, Tewessteg 4, 20249 Hamburg, Germany
[2]European Space Agency - ESRIN, Via Galileo Galilei, Casella Postale 64, 00044 Frascati (Roma), Italy

*Correspondence to:* Thomas Kaminski (Thomas.Kaminski@Inversion-Lab.com)

**Abstract.** The vehicles that fly the satellite into a model of the Earth System are *observation operators*. They provide the link between the quantities simulated by the model and quantities observed from space, either directly (spectral radiance) or indirectly estimated through a retrieval scheme (biogeophysical variables). By doing so, observation operators enable modellers to properly compare, evaluate and constrain their models with the model-analogue of the satellite observations. This paper provides the formalism and a few examples of how observation operators can be used, in combination with data assimilation techniques, to better ingest satellite products in a manner consistent with the dynamics of the Earth System expressed by models. It describes commonalities and potential synergies between assimilation and classical retrievals. This paper explains how the combination of observation operators and their derivatives (linearisations) form powerful research tools. It introduces a technique called automatic differentiation that greatly simplifies both development and maintenance of code for evaluation of derivatives. Throughout this paper, as special focus lies on applications to the carbon cycle.

## 1 Introduction

Earth System Models (ESMs) are complex software capturing our knowledge of how the ocean, atmosphere, land and ice operate and interact. ESMs provide scientists with powerful tools to better understand our global environment, its evolution, and the potential impact of human activities (e.g. analyses of relevant processes, their interaction and feedback mechanisms). ESM applications range from numerical weather prediction (NWP) to seasonal to decadal forecasting (see, e.g., Stockdale

et al., 2011; Smith et al., 2013) to climate projections on centennial (Pachauri et al., 2014) or even longer (Jungclaus et al., 2010) scales.

Before being used for predictions, ESMs and their components should be confronted with observations in order to assure their realism (validation). Such validation procedures can be extended to standardised assessments of model performance in so-called benchmarking systems by evaluation of a set of observation-based metrics (see, e.g., Blyth et al., 2011; Luo et al., 2012). This involves the definition of metrics that quantify the model performance through the fit to observations. A further step towards the rigorous use of the observations is their ingestion in formal data assimilation procedures, e.g. to constrain the model's initial state (initialisation) or tunable parameters in the model's process representations (calibration).

Such confrontation with observations is hampered by the fact that observed and modelled quantities typically differ in nature or scale (in space and time). For example, a flask sample of the atmospheric carbon dioxide concentration provides a value at a specific point in space and time, whereas an atmospheric tracer model operates in a discretised representation of space and time, i.e. on values that refer to a box in the atmosphere and a particular period of time. Any comparison of the two quantities (modelled and observed) must, hence, take the uncertainty arising from this representation error into account (see, e.g., Heimann and Kaminski, 1999). Another example is a vertical profile of the ocean temperature and salinity provided by a floating buoy (see, e.g., http://www.argo.ucsd.edu). Again the spatial scales of the observation and the model do not match (in the horizontal dimension). In addition, ocean models are formulated in terms of potential temperature rather than temperature. Since we can only compare quantities of the same nature, some form of transformation is required before any comparison can take place. Such a difference in nature is intrinsic to observations from space, where the raw quantities measured by satellites, i.e. photon counts (Mathieu and O'Neill, 2008), are by nature only indirectly (through radiative transfer processes) related to the model quantities of interest.

The link from the model to the observations is provided through a set of relationships expressed in terms of an *observation operator*. We can think of an observation operator as an arm, which enables the ESM to access a particular type of observation (see Figure 1). We stress that the usage of the term *operator* is not meant to imply the linearity of the observation operator. In fact observation operators are ranging in complexity from a simple interpolation or integral scheme up to a chain of sophisticated non-linear radiative transfer models.

The layout of the remainder of this paper is as follows. Section 2 introduces the concept of an observation operator and presents examples. The role of observation operators in applications is presented in section 3. Section 4 highlights the use of derivatives of observation operators and introduces *automatic differentiation*, a technique to provide these derivatives. Finally, section 5 draws conclusions.

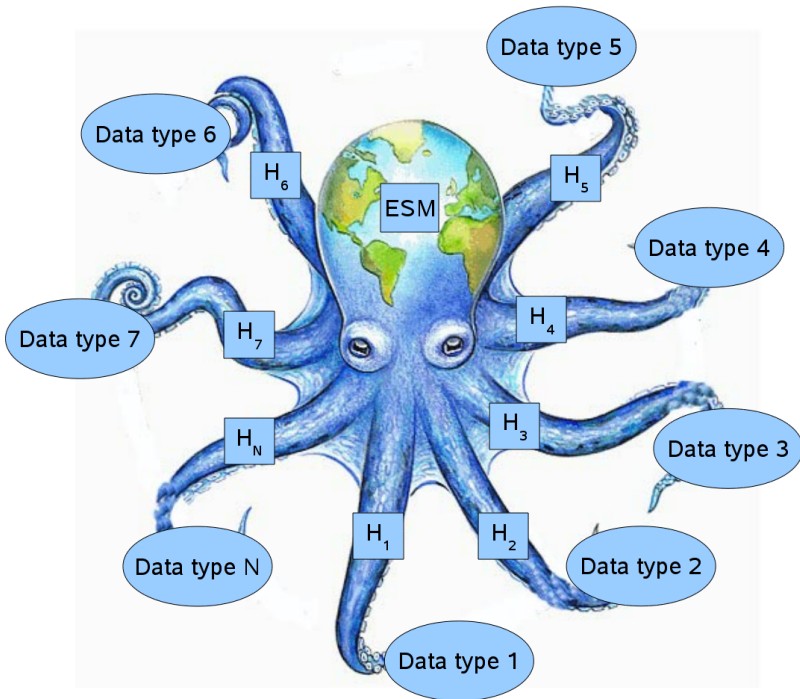

**Figure 1.** Schematic of an ESM assessing several of data types via observation operators $H_1$, ..., $H_n$.

## 2 Observation Operators

### 2.1 Definition

Mathematically the observation operator is defined as a mapping $H$ from the vector of state variables $\mathbf{z}$ (of the model) onto the vector of observations $\mathbf{y}$:

$$H : \mathbf{z} \mapsto \mathbf{y} \tag{1}$$


The observation vector can include, for example, observed radiances, radar backscatter, or in situ observations. The vector of the model's state variables (*state vector*) defines the simulated system for a given time step at all points in space, and the evolution of the system is described by a sequence of state vectors, forming a trajectory through the state space. We note that equation (1) may be generalised in the sense that the simulation of observations of temporal averages or integrals (e.g. an increment in above ground biomass over several years, or an albedo covering several weeks) may require not only a single state but the trajectory over the averaging/integration interval. The state variables of a dynamical model are also called prognostic variables, to contrast them with diagnostic


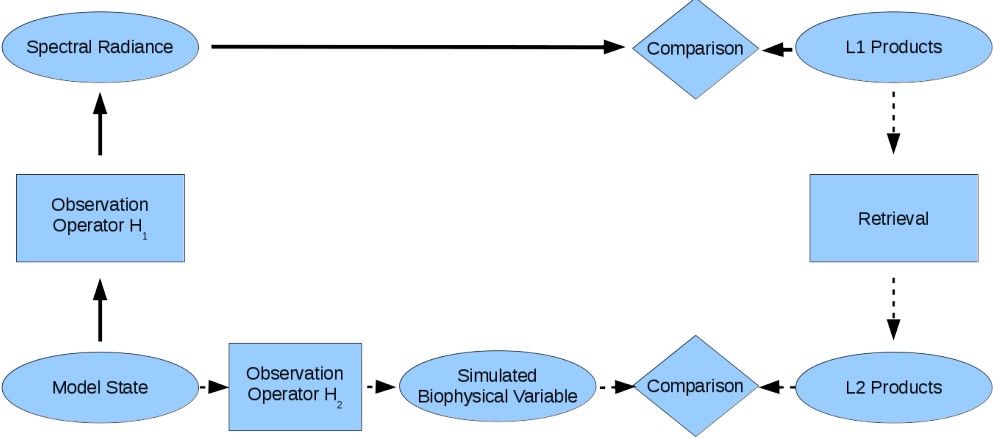

**Figure 2.** Model-Data Comparison at the sensor level (Level 1, solid arrows) and at the level of geophysical variables (Level 2, dotted arrows). Ovals denote data, rectangulars some form of processing.

variables, which are computed from the state and evolve only indirectly through the evolution of the
state. For example the albedo of the land surface is diagnosed from the state of the vegetation-soil
system. Hence, if we achieve a change in the model state at any given point in time, the model will
then propagate this change of state forward in time, and we achieve a change of the model trajectory
(e.g. to improve the fit to observations). This means, to bring observational information into the
model, we must link the observations to the state: In other words the model's state vector constitutes
the interface between the model and the observation operator.

The solid path in Figure 2 sketches how an observation operator ($H_1$) enables the comparison of
simulated and observed values at the sensor level, i.e. at the level of spectral radiances, typically
referred to as level 1 data products (Arvidson et al., 1986). Another way to make Earth Observa-
tion (EO) data accessible to dynamical models is by deriving, by means of a retrieval algorithm, a
bio-geophysical variable (in the following just denoted as geophysical variable) from the satellite ob-
servations. Such EO products are usually called level 2 data products (and level 3 refers to products
on a space-time grid). Internally, the retrieval algorithm also relies on a functional relationship that
maps the geophysical variable(s) of interest onto the spectral radiance. This mapping is similar, if not
identical to the observation operator $H_1$, although the term used by the EO community is *forward
model*. The retrieval can be regarded as an inversion of $H_1$. As the examples below will illustrate,
the retrieved level 2 product will typically not exactly coincide with a component of the model state
vector. Hence, the confrontation of level 2 data with the model (dotted path in figure 2) also requires
an observation operator (denoted by $H_2$).

## 2.2 Examples

Figure 3 attempts to sketch a generic observation operator $H_1$, which links a model's state vector to observed spectral radiance. For the sake of clarity the figure focuses on processing steps that map one variable onto another and omits further important steps that involve transformations in space and time, i.e. interpolation, averaging, or orbit simulation.

The simulation of spectral radiances at the sensor level requires information from the atmosphere
and the land/ocean surface, including the description of ice or snow covers. Hence, the observation operator typically consists of various modules. First, from the model state the relevant electromagnetic signatures are simulated. For example, for a passive optical sensor observing the terrestrial vegetation this would be the reflected sun light, and it would be computed by a model of the radiative transfer within the canopy, for examples see, e.g., Pinty et al. (2006) or Loew et al. (2014). For a
passive microwave sensor that observes sea ice and snow, this would be the thermal emission, and it would involve a model of the radiative transfer within the sea ice and snow pack (see, e.g., Wiesmann and Mätzler, 1999; Tonboe et al., 2006). In the atmosphere this could be a model for the emissivity of clouds as a function of the atmospheric state. The next step covers the path through the atmosphere from the observed components to the sensor and requires a model of the radiative transfer through
the atmosphere. Prime examples are the Radiative Transfer for Tiros Operational Vertical Sounder (RTTOV, Eyre, 1991; Saunders et al., 1999) for the microwave and infra-red domain, 6S (Vermote et al., 1997) for the solar domain, or the Optimal Spectral Sampling (OSS) method (Moncet et al., 2008). The output of the radiative transfer model can be compared with a level 1 product.

Each type of observation requires its own observation operator in order to be accessible to models.
The complexity of the observation operator typically reflects a compromise between the accuracy required for the application at hand and the available computational resources. In a space mission, the observation operator depends on characteristics such as the geometry of the observation (as a function of the orbit of the platform) or the measuring principle and, thus, spectral sensitivity of the sensor. The observation operator also depends on the formulation of the dynamical model. One
aspect is the state space, which depends on the model formulation. For example, an atmospheric model can either diagnose clouds or include them in the state space (Chevallier et al., 2004). In the former case the diagnostic cloud model is part of the observation operator in the latter it is not. Even though parts of an observation operator are usually model-dependent, it is desirable to implement the observation operator in a modular form with carefully designed interfaces. This modularity max-
imises the flexible use and reuse for assimilation and retrievals and the adaptation to new models or observations, i.e. it ensures multi-functionality.

The crucial role of observation operators is reflected in comparison exercises such as the radiation transfer model intercomparison (RAMI) initiative for the transfer of radiation in plant canopies and over soil surfaces (Pinty et al., 2001; Widlowski et al., 2007, 2013, 2015). A similar activity for
the atmosphere is the international Intercomparison of 3D Radiation Codes (I3RC) project (Caha-

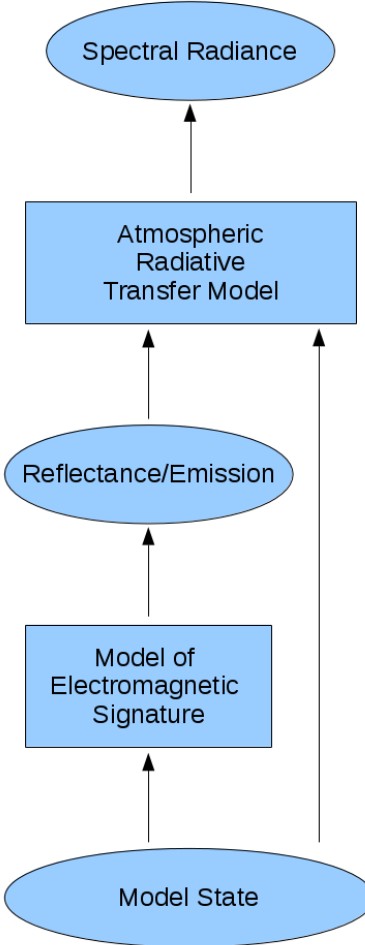

**Figure 3.** Generic scheme of an observation operator for spectral radiance. Oval boxes denote data, rectangular boxes denote processing.

lan et al., 2005). The I3RC focuses on the interaction of solar and thermal radiation with cloudy atmospheres. Another activity in this domain is the Cloud Feedback Model Intercomparison Project (CFMIP) which has set up the CFMIP Observation Simulator Package (COSP) (Bodas-Salcedo, 2011): The modular package includes a set of observation operators that map model output consisting of "vertical profiles of temperature, humidity, hydrometeor (clouds and precipitation) mixing ratios, cloud optical thickness and emissivity, along with surface temperature and emissivity" onto a set of level 2 products retrieved from "the following instruments: CloudSat radar, CALIPSO lidar, ISCCP, the MISR, and the Moderate Resolution Imaging Spectroradiometer (MODIS)". The above-mentioned "fast radiative transfer code RTTOV can also be linked to COSP to produce clear-sky brightness temperatures for many different channels of past and current infrared and passive mi-

crowave radiometers." Not only does COSP greatly simplify the comparison of model output with EO products. Using standardised interfaces allows the comparison of multiple models through the same observation operators with EO data from various sources, and thus facilitates the attribution of a model-data mismatch to aspects of the model, the observation operator, or the observations. The Community Microwave Emission Modelling Platform (CMEM, Drusch et al., 2009) takes a similar role for the modelling of the emissivity of the canopy-soil system in the spectral domain from 1 to 20 GHz. For example, de Rosnay et al. (2009) use 12 different configurations of the modular system in their Land Surface Models Intercomparison Project (ALMIP).

## 3 Applications of Observation Operators

This section starts with an introduction of the formalism behind advanced data assimilation and retrieval schemes. The details of the formalism are useful to understand the application examples in this section (and the commonalities between assimilation and retrievals) and the need for derivative information that is discussed in section 4.

### 3.1 Formalism of Data Assimilation and Retrieval

Data assimilation is a procedure to combine the information from observations with the information in a dynamical model. There is a range of data assimilation techniques with varying degree of sophistication. The simplest techniques try and replace a component of the model state vector by an observation, or, more precisely, some average of the two. More advanced approaches can assimilate observations $\mathbf{y}$ which are linked to the state through an observation operator $H$. $H$ can be an observation operator for in situ data or for EO data, for example the operators $H_1$ and $H_2$ introduced in section 2 (see equation (1) and figure 3)). The assimilation problem is typically formalised as a minimisation problem for a misfit function

$$J(\mathbf{x}) = \frac{1}{2}\left(H(\mathbf{x}) - \mathbf{y}\right)^T \mathbf{U_y}^{-1}\left(H(\mathbf{x}) - \mathbf{y}\right) + \frac{1}{2}\left(\mathbf{x} - \mathbf{x_{pr}}\right)^T \mathbf{U_{xpr}}^{-1}\left(\mathbf{x} - \mathbf{x_{pr}}\right) \ , \tag{2}$$

where $\mathbf{x}$ denotes the vector of unknowns. Even though in some applications this vector of unknowns may coincide with the model state, $\mathbf{z}$, this is not generally the case (as will be discussed below), and we need to make a clear distinction between both objects.

The function $J(x)$ is composed of two terms. The first term quantifies the misfit between the observations and their simulated counterpart (observational term). $\mathbf{U_y}$ has to account for the uncertainty in the observations and the uncertainty imposed by imperfection of the model, including the above-mentioned representativeness in space and time. For diagonal $\mathbf{U_y}$ (uncorrelated uncertainty) it reduces to a least squares fit to the observations. The second term quantifies the deviation of the model state from the prior information $\mathbf{x_{pr}}$ (prior term, often also called background). This term provides a means to include information in addition to the observational information into the assimilation procedure and it ensures the existence of a minimum in cases where the observational

information is not sufficient to constrain the unknowns. Both terms, observation misfit and prior, are weighted in inverse proportion to the respective uncertainties, i.e. the combined uncertainty in the observations and observation operator, $\mathbf{U_y}$, and the uncertainty in the prior information, $\mathbf{U_{xpr}}$. The superscript $T$ denotes transposition. Note that the equation does not require the observations to be provided in the space time grid of the model. The observations can come in any spatio-temporal distribution, e.g. the above mentioned point measurements or orbits, as long as we can formulate the appropriate observation operator.

Equation (2) formalises what in numerical weather prediction is called three dimensional variational assimilation (3D-Var, Courtier et al., 1998), more precisely its analysis step, which is then followed by a forecast step. Operationally the assimilation scheme is run in cyclic mode through these two steps. In such a cyclic scheme, the prior information is provided by the previous forecast, i.e. it is consistent with the dynamical information from the model, and at the same time suffers from errors in the model. In this setup, the specification of the prior uncertainty is particularly challenging (see, e.g., Bannister, 2008a, b).

The model dynamics are even more emphasised when the scheme of equation (2) is extended to contain observations $\mathbf{y_i}$ at different time steps ($i =, 1...,n$) to constrain the initial state $\mathbf{z_0} = \mathbf{x}$ through:

$$J(\mathbf{x}) = \frac{1}{2} \sum_{i=1,n} \left( H(\mathbf{z_i}(\mathbf{x})) - \mathbf{y_i} \right)^T {\mathbf{U_{y_i}}}^{-1} \left( H(\mathbf{z_i}(\mathbf{x})) - \mathbf{y_i} \right) + \frac{1}{2} \left( \mathbf{x} - \mathbf{x_{pr}} \right)^T {\mathbf{U_{xpr}}}^{-1} \left( \mathbf{x} - \mathbf{x_{pr}} \right). \quad (3)$$

This is the setup of the analysis step in four dimensional variational assimilation (4D-Var) schemes, where the dynamical model $M$ is used as a constraint that links the states at all observation times via

$$\mathbf{z_{i+1}} = \mathbf{M}(\mathbf{z_i}) \tag{4}$$

to the initial state $\mathbf{z_0} = \mathbf{x}$. For convenience the notation suppresses the time dependent nature of $H$ and $M$, and it also assumes that the data uncertainties at different time steps are uncorrelated. While 4D-Var solves a single minimisation problem to find a (dynamically consistent) model trajectory, 3D-Var is a sequential approach, i.e. it solves a sequence of minimisation problems, which yield a dynamically inconsistent sequence of model states. In operational NWP, as mentioned above, the application of 4D-Var is in a cyclic procedure, i.e. also in a sequential manner. Section 3.2 will describe applications where this is not the case.

In the 4D-Var approach, the vector of unknowns $\mathbf{x}$ can be extended from the initial state to boundary values and process parameters (model calibration). Since these are external controls to the dynamical system, $\mathbf{x}$ is also called control vector, a term taken from control theory (Lions, 1971). We usually try to select the control vector such that it comprises the fundamental unknowns of the system at hand, i.e. those with the highest uncertainty (Kaminski et al., 2012b). An extension of the 4D-Var approach (weak constraint 4D-Var) allows deviations from the model trajectory, which are included as additional components into the control vector (see, e.g., Zupanski, 1997).

The *Kalman Filter* is another sequential approach. Its analysis step solves a slightly simplified form of equation (2), in which $H$ is replaced by its linearisation $\mathbf{H}$ (Jacobian matrix) around the prior. This allows an analytic solution $\mathbf{x_{po}}$ of equation (2):

$$\mathbf{x_{po}} = \mathbf{x_{pr}} - \mathbf{U_{xpo}}\mathbf{H^T}\mathbf{U_y}^{-1}\left(\mathbf{Hx_{pr}} - \mathbf{y_i}\right) \tag{5}$$

the evaluation of which involves the inversion of the matrix

$$\mathbf{U_{xpo}} = (\mathbf{H^T}\mathbf{U_y}^{-1}\mathbf{H} + \mathbf{U_{xpr}}^{-1})^{-1} \tag{6}$$

which is typically of high dimension (e.g., $10^7$ in NWP) and expresses the uncertainty range in $\mathbf{x_{po}}$ that is consistent with uncertainty ranges in the data and the prior values.

In case of linear $H$ and Gaussian probability densities for the prior and the data, the solution of equation (2) is Gaussian as well and, thus, completely described by its mean (equation (5)) and covariance (equation (6)). This formalism is, for example, applied in inverse modelling of the atmospheric transport of carbon dioxide (Enting, 2002), where an atmospheric transport model takes the role of $H$, and the space-time distribution of the surface fluxes takes the role of $\mathbf{x}$. Note that the cost function's second derivative (Hessian matrix) $\mathbf{J}''$ is related to $\mathbf{U_{xpo}}$ through:

$$\mathbf{U_{xpo}} = \mathbf{J}''(\mathbf{x})^{-1} \tag{7}$$

In the non-linear case (i.e. $H$ or $M$ are non-linear) we cannot solve equation (2) or equation (3) analytically, but via the cost function's Hessian we can use equation (7) to approximate $\mathbf{U_{xpo}}$. Via a linearisation $\mathbf{N}$ of the model that links the control variables to model outputs of interest $f$ we can approximate the uncertainty range of these model outputs $\mathbf{U_f}$ by:

$$\mathbf{U_f} = \mathbf{N}\mathbf{U_{xpo}}\mathbf{N^T} \tag{8}$$

The alternative to the above assimilation approaches (which are based on linearisations) are ensemble methods such as Markov Chain Monte Carlo (see, e.g., Metropolis et al., 1953), Ensemble Kalman Filter (Evensen, 2003) techniques, or particle filters (see, e.g., van Leeuwen, 2009), which rely on forward simulations to sample the control space. The feasible ensemble size is limited by the computational demands which are essentially determined by the complexity of the underlying model. We also note the challange of filter degeneracy that limits the applicability of particle filters to high-dimensional problems (see, e.g., Snyder et al., 2008).

We used equation (2) to introduce the formalism of data assimilation. The *same* equation also plays a central role in retrievals. Minimisation of equation (2) describes a retrieval algorithm for the entire state. The prior term regularises what is otherwise an underdetermined inverse problem: Several of the unknown variables that influence the observed signal vary continuously with altitude (continuous vertical profiles). Even though we formulate our observation operators on a vertical grid, there are typically "fewer" measurements than unknowns. Consequently there are many sets of unknown variables that yield an equal fit to the observations, i.e. we have to deal with a non-zero null space (Tarantola, 2005). The word fewer was put in quotation marks to indicate that, more

precisely, it is not only the ratio of the numbers of observations to unknowns that matters here, it is also the capability of the observations to constrain the unknowns. The null space is the sub space of the control space that is not constrained by the observations, and often including more observations of the same type does not help to reduce the dimension of the null space. The prior term provides additional information on every unknown and helps the retrieval algorithm to find a unique solution.

Further, equation (6) or equation (7) are used to furnish the retrievals with uncertainty ranges.

Another perspective on the assimilation of level 1 data is to regard it as an advanced form of retrieval, and the assimilation system as an advanced retrieval algorithm that optimally combines the information from remote sensing, radiative transfer and dynamical model. The other point to note is that $H$ is usually not constant in space and time. For example the radiative transfer in the optical

domain is affected by atmospheric water vapour and aerosols. A retrieval of, say, a land surface variable requires information on clouds and aerosols. In a coupled atmosphere-land model these are available in a form that is dynamically consistent with the state of the land surface but, on the other hand, also affected by errors in the model.

### 3.2 Data Assimilation examples

The prime example of an atmospheric 4D-Var system is the one (Rabier et al., 2000) operated at the European Centre of Medium-Range Weather Forecasts (ECMWF) in their Integrated Forecasting System (IFS, Courtier et al., 1998). The 4D-Var system is in operation since 2003; meanwhile most of the assimilated observations are remotely sensed radiances. Observations are provided by about 50 different sensors and used with appropriate observation operators. The system is used with a 12 hour

assimilation window to initialise the operational forecast. Several other weather services (including those of Canada, France, and the UK) are running similar 4D-Var systems. A recent development at NWP centres are hybrid approaches that combine ensemble and variational approaches. Such a hybrid approach is operational, e.g., at ECMWF (Buizza et al., 2008; Isaksen et al., 2010; Bonavita et al., 2012) or the NWP centres of the UK Clayton et al. (2013) or Canada (Buehner et al., 2010).

A prominent example of a variational ocean assimilation system was set up by the ECCO consortium (see http://www.ecco-group.org) around the MITgcm (Marshall et al., 1997). The system (Stammer et al., 2002) uses a combination of in-situ observations and level 2/3 remote sensing products (including sea surface height, sea surface temperature, wind-stress, and geoid) for *ocean state estimation* over decadal-scale assimilation windows (Wunsch and Heimbach, 2006). Owing to these

long assimilation windows the prescribed exchange fluxes with the atmosphere are a major source of uncertainty in their model trajectory. Hence, this boundary condition is included in the control vector along with the initial state. Various applications of the assimilation product require closed property budgets over the entire assimilation window which are achieved via variational approaches in contrast to sequential approaches. Examples are mechanistic or diagnostic studies of climate variability

or oceanic tracer transport problems (Wunsch et al., 2009).

A recent example of a regional variational assimilation system for the coupled ocean sea-ice system in the Northern latitudes was developed by Kauker et al. (2015). This system is operated for assimilation windows from a few weeks to a few years. Their control vector combines (depending on the application) the initial state, boundary conditions, and process parameters. The system is constrained by hydrographic in-situ observations and level 2/3 products of sea surface temperature, sea ice concentration, thickness, and displacement.

An example for the global terrestrial vegetation is provided by the Carbon Cycle Data Assimilation System (CCDAS). Initially set up for the assimilation of in-situ observations of the atmospheric $CO_2$ concentration (Rayner et al., 2005), the system was extended step-by-step with observation operators for several level2/3 products, namely Fraction of Absorbed Photosynthetically Active Radiation (FAPAR) products (Knorr et al., 2010; Kaminski et al., 2012a), the column-integrated atmospheric carbon dioxide concentration (XCO2) (Kaminski et al., 2010, 2016b), and the surface layer soil moisture (Scholze et al., 2016). The observation operator for FAPAR was a considerable extension to the previous system, because it required modules for the simulation of vegetation phenology and of hydrology, which were previously provided by an off-line calculation. The observation operators for $CO_2$ and XCO2 include models of the atmospheric transport that solve the continuity equation for carbon dioxide (Heimann, 1995; Heimann and Körner, 2003). The observation operator for surface layer soil moisture was derived by modification of the initial bucket formulation of the soil hydrology model (which had no equivalent to the thin surface layer that is observed from space). The assimilation window ranges from years to decades. Considering uncertain values of the parameters (constants) in process formulations as the major source of uncertainty in the model trajectory, the control vector is composed of (depending on the setup) in the order of 50-100 (in extreme cases up to 1000) process parameter values. This type of application is called parameter estimation or model calibration.

### 3.3 Retrieval Examples

The integrated retrieval of Toudal (1994) or Melsheimer et al. (2009) solves simultaneously for geophysical variables (level 2 data) of the atmosphere (wind speed, total water vapour, cloud liquid water) the ocean (sea surface temperature) and the sea ice (ice surface temperature, total sea ice concentration, multi-year ice fraction). Technically, as $H$ is non-linear, they use equation (5) in an iterative procedure ($\mathbf{x_{po}}$ from one step is provided as $\mathbf{x_{pr}}$ to the subsequent step), which recomputes $\mathbf{H}$ by linearisation around the current $\mathbf{x_{pr}}$. Upon convergence they deliver posterior uncertainties via equation (6). Their input are radiances (brightness temperatures) observed by the Advanced Microwave Scanning Radiometer for EOS (AMSR-E). Their prior values are taken from a range of sources including analysis data provided by an NWP assimilation system, or separate univariate retrievals. This integrated retrieval is performed individually for each observed point in space and time, at 12.5 km horizontal resolution. Although the use of the same level 1 data in an assimilation system

(ensuring dynamical consistency between the atmosphere, ocean and sea ice components) appears desirable, it is highly challenging in various respects: From a software development perspective, because it would require an assimilation system built around a coupled atmosphere, ocean and sea ice model. From a computational perspective, because a single run of the coupled model at 12.5 km resolution is already computationally expensive, let alone an iterative assimilation scheme.

An example for the land surface is the Joint Research Centre-Twostream Inversion Package (JRC-TIP) (Pinty et al., 2007), which solves equation (2) for model parameters controlling the radiation transfer regime in vegetation canopies, namely the effective Leaf Area Index (LAI) and the spectral scattering properties of the vegetation and the soil. The latter information is then used to compute the spectral fluxes scattered by, absorbed in and transmitted through the vegetation layer as well the fluxes absorbed in the background (radiant fluxes). Further, the system uses equation (7) to infer the uncertainty in the retrieved parameters and equation (8) to propagate these forward to uncertainties in the simulated radiant fluxes. In its typical setup the system uses observed albedos in two broad wavebands (visible and near infrared) (Pinty et al., 2007, 2011a, b). The JRC-TIP is constructed around a one dimensional two stream model, which takes three-dimensional radiative transport effects into account (Pinty et al., 2006). As a consequence the retrieved vegetation parameters are effective parameters (i.e. their values are only meaningful within this model) and are determined such that the radiant fluxes are simulated as accurately as possible. This illustrates a crucial point when confronting retrieved level 2 variables with their ESM counterparts: It is essential that the variables have the same definition in the forward model that is used for the retrieval and in the observation operator that is used for its assimilation. For the JRC-TIP products this is the case for the radiant fluxes and soil parameters, for the effective vegetation parameters it requires the use of the same two-stream model in the observation operator. An in-depth description of JRC-TIP, its use for generation of EO products, and the validation of these products is provided by a dedicated contribution to this special issue (Kaminski et al., 2016a).

The Earth Observation Land Data Assimilation System (EO-LDAS, Lewis et al., 2012) is a retrieval and a data assimilation framework for various types of EO data. Primarily it uses a weak-constraint variational approach, i.e. the weak constraint form of equation (3), in combination with a simple model of the land surface dynamics (e.g. persistence), which acts as a regularisation in the spatial and temporal domains. The system can, however, also perform single pixel retrievals or be operated in a sequential manner. EO-LDAS development started for the optical domain with the semi-discrete model of the vegetation canopy (Gobron et al., 1997) as observation operator. The system is being extended by further observation operators, including the above mentioned CMEM for passive microwave observations. This means it has the capability to simultaneously use EO over a range of spectral domains and exploit their complementarity. To achieve the computational performance that it necessary for global scale processing of high-resolution EO data, the system is also operated (Gómez-Dans et al., 2016) with fast approximations of RT models (including the above

mentioned atmospheric RT code 6S) by so-called emulators. A similar regularisation strategy in the

350 temporal domain was also presented by Lauvernet et al. (2008), who operated a variational inversion scheme around a chain of coupled RT models, i.e. PROSPECT (Jacquemoud and Baret, 1990) for the leaf optical properties, SAIL (Verhoef, 1984) for the canopy RT, and SMAC (Rahman and Dedieu, 1994) the atmospheric RT.

A serious practical difficulty in data assimilation is the specification of $\mathbf{U_y}$. This is particulary

relevant in multi-data stream assimilation, because $\mathbf{U_y}$ defines the respective weights of the data streams. In the case of level 2 data $\mathbf{U_y}$ is the posterior uncertainty of the retrieval. Since the retrieval is typically carried out point by point, uncertainty correlations in space and time are difficult to assess. Another issue is the data volume required by the uncertainty information: For a product of a retrieved geophysical variable at $n$ points in space and time $\mathbf{U_y}$ contains (taking its symmetry into

account) $n * (n + 1)/2$ different values, a volume that is usually prohibitive in real world applications. The challenge is to develop ways of providing (approximations of) $\mathbf{U_y}$ that retain the essential information in a minimal data volume. One approach is through appropriate variable transformations based on a singular value decomposition of the observation operator (Joiner and Da Silva, 1998; Migliorini, 2012). Another approach is a parametric model of $\mathbf{U_y}$ as, for example, provided

by (Reuter et al., 2016) for the XCO2 product retrieved from SCIAMACHY observations through their Bremen Optimal Estimation DOAS (BESD, Reuter et al., 2011) algorithm. Generally, the contribution by the observational uncertainty $\mathbf{U_y}$ is certainly easier to specify in the case of level 1 data. Their direct assimilation automatically propagates, through the observation operator $H_1$, the full information content of $\mathbf{U_y}$ into the model.

A related topic is the consistency of the prior information ($\mathbf{x_{pr}}$ and $\mathbf{U_{xpr}}$ in equation (2)) used in the retrieval scheme with the scheme that subsequently uses these retrievals in a dynamical model, e.g. for assimilation. Rodgers and Connor (2003) demonstrate, how the $\mathbf{x_{pr}}$ used in the retrieval can be replaced with that simulated by the dynamical model, provided that $\mathbf{x_{pr}}$ and the so-called resolution operator (Backus and Gilbert, 1968) $\mathbf{U_{xpo}H^T U_y}^{-1}\mathbf{H}$ of the retrieval (also called aver-

aging kernel), i.e. equation (5), are available. Chevallier (2015) goes one step further and addresses remaining inconsistencies in $\mathbf{U_{xpr}}$ and highlights their effect in his atmospheric transport inversion using retrievals of XCO2. More generally, Migliorini (2012) derives requirements on the retrieval, such that assimilation of the retrieved level 2 products is equivalent to direct assimilation of level 1 products.

### 3.4 Observing System Simulation Experiments and Quantitative Network Design

Observing System Simulation Experiments (OSSEs) and Quantitative Network Design (QND) are two methodologies that evaluate observation impact on assimilation systems. By an observing system or observational network we understand the superset of all observations that are made available

to an assimilation system. We only give a brief introduction to the topic, as QND for the carbon cycle is addressed by another contribution to this special issue.

An OSSE (see, e.g., Böttger et al., 2004; Masutani et al., 2010; Timmermans et al., 2015) uses a model plus observation operators to simulate in a model analogues of observations that would be collected by a potential observing system (often the current observing system extended by a potential new data stream). The model is also used to simulate, in a so-called "nature run", a surrogate of reality, i.e. a reference trajectory over the period of investigation. Then an assimilation/forecast system (preferably built around a different model) is used to evaluate some measure of the performance of the potential observing system and its sub-systems. In NWP, the performance of an observing system is usually quantified by the quality (skill) of a forecast from the initial value that was constrained by the observation system. Via this procedure one can, for example, assess the added value of a planned mission in terms of an increment in forecast skill. We also note a related approach, Observing System Experiments (OSEs), which assess observation impact by removing one or several *existing* data streams from the list of observations used in a data assimilation system.

QND (for an overview see Kaminski and Rayner (2008)) relies on the ability of an assimilation system to evaluate posterior uncertainties on target quantities of interest via equation (7) and equation (8). For a linear model, this propagation of uncertainty is independent of the observational value, it just depends (via equation (7) and equation (3)) on the data and prior uncertainties, the sensitivity of the observations with respect to the control variables and (via equation (8)) on the sensitivity of the target quantity $f$ to the control variables. A first application to mission design was presented by Rayner and O'Brien (2001), who ran an atmospheric transport inversion system built around a linear model of the atmospheric transport of carbon dioxide in QND mode. They assessed the utility of remotely sensed carbon dioxide in constraining its surface fluxes. Their benchmark was the in-situ flask sampling network. Kaminski et al. (2010) generalised the method to the above mentioned CCDAS, and assessed the utility of XCO2 observations by an active LIDAR instrument. The performance of the observing system is quantified by posterior uncertainty of surface fluxes and compared to the performance of the in-situ network. Kaminski et al. (2012a) use CCDAS to assess the performance of potential optical sensor configurations in constraining the vegetation's carbon and water fluxes. Their benchmark was the MERIS sensor. Another QND application (Kaminski et al., 2015) evaluated airborne sampling strategies for sea ice thickness and snow depth in the Arctic using simultaneous laser altimeter and snow radar observations.

For both approaches, OSSE and QND, the importance of suitable observation operators is obvious. A disadvantage is that the result depends on the model. Both techniques require the specification of data uncertainties for the hypothetical data streams to be evaluated.

## 4 Derivatives of Observation Operators

This section first summarises how the capability to evaluate derivatives of the observation operator is used in efficient schemes for retrieval, assimilation or QND and then introduces a technique for providing derivative information.

In variational assimilation, equation (2) or equation (3) are typically minimised in an iterative procedure that varies $\mathbf{x}$. To do this efficiently even for high-dimensional control spaces, so-called gradient algorithms are employed. They rely on the capability of evaluating the gradient of $J$ with respect to $\mathbf{x}$ to define a search direction in the space of unknowns. The gradient is useful, because it yields the direction of steepest ascent. For $J(x)$ of equation (2) straight-forward differentiation with respect to $\mathbf{x}$ yields (see, e.g., section 3.4.4 of Tarantola (2005)) the gradient

$$\nabla J(\mathbf{x}) = \mathbf{H}(\mathbf{x})^{\mathbf{T}} \mathbf{U_y}^{-1} \left(\mathbf{H}(\mathbf{x}) - \mathbf{y}\right) + \mathbf{U_{xpr}}^{-1} \left(\mathbf{x} - \mathbf{x_{pr}}\right), \tag{9}$$

and we see that its evaluation requires the capability to multiply the transpose of $\mathbf{H}$ with a vector. The uncertainty estimation via equation (7) based on $\mathbf{J}''$ requires, in addition, second derivative information on $H$. This second derivative expresses the curvature of (the components) of $H$, i.e. the change of its linearisation corresponding to a unit change of $\mathbf{x}$.

Likewise the Kalman filter requires derivatives of $H$: In equation (5) it multiplies the matrix $\mathbf{H}$ and its transpose with vectors, and for the evaluation of equation (6) it needs to invert a matrix that contains $\mathbf{H}$ and its transpose. One can do this inversion by precomputing $\mathbf{H}$ or by so-called matrix-free methods that repeatedly multiply $\mathbf{H}$ and its transpose with vectors.

As mentioned, advanced retrieval algorithms are based on the same equations, i.e. they typically solve equation (2) either via gradient methods, or (possibly repeated) application of equation (5), and use either equation (7) or its approximation equation (6) to estimate the posterior uncertainty. Hence, they benefit in the same way from derivatives of $H$ as data assimilation systems. The same holds for QND schemes, which are based on the computation of uncertainties via equation (7) or its approximation, equation (6).

Traditionally derivatives were approximated by multiple forward runs (finite difference approximation) (see, e.g., Toudal, 1994; Melsheimer et al., 2009; Govaerts et al., 2010; Dubovik et al., 2011). This discretised procedure has two disadvantages: The first is the limited accuracy of this gradient approximation (providing only the linear term of the Taylor series), which degrades the performance of the above listed algorithms. For example, incorrect gradient information will slow down or prematurely stop the iterative minimisation of $J$, because gradient-based minimisation algorithms rely on the consistency of evaluations of $J$ and its gradient. The other disadvantage is that the computational cost of this approximation grows linearly with the length of the control vector.

Both disadvantages can be avoided by Automatic differentiation (AD, Griewank, 1989). AD is a procedure which generates source code for evaluation of derivatives from the code of the underlying function. In the current case this function is the observation operator mapping the state variables

onto remote sensing products. The function code is decomposed into elementary functions (such as $+, -, \sin(\cdot)$), for which the derivative (local Jacobian) is straightforward to derive. The derivative of the composite function is then constructed via the chain rule as the product of all local Jacobians. According to the associative law, this multiple matrix product can be evaluated in arbitrary order without changing the result. The *tangent linear* code (or just tangent code) does this evaluation in the same order as the function is evaluated, which is called forward mode of automatic differentiation. The *adjoint* code uses exactly the opposite order, which is called reverse mode of automatic differentiation. Even though both modes yield the same derivative, depending on the dimensions of the function to be differentiated, there may be large differences in their computational efficiency: The CPU time required by tangent code is proportional to the number of the function's input variables but independent on the number of output variables. By contrast, the CPU time required by the adjoint code is proportional to the number of output variables and independent of the number of input variables. Both the tangent and adjoint codes use values from the function evaluation (*required values* (see, e.g., Giering and Kaminski, 1998; Hascoët et al., 2004). Providing required values to the adjoint code is more complicated than to the tangent code. Being an application of the chain rule, AD provides derivatives that are accurate up to rounding error.

For variational assimilation we require the derivative of the scalar-valued cost function $J(x)$ of equation (2) or equation (3) with respect to a usually high-dimensional vector $\mathbf{x}$. For a state-of-the-art model, only the adjoint can provide this derivative with sufficient efficiency. A product $\mathbf{H}v$ of $\mathbf{H}$ with a vector $v$ yields the directional derivative of $\mathbf{H}$ in the direction defined by $v$, i.e. the derivative of the function $H(x + tv)$ of a scalar unknown $t$. Hence, this type of product is evaluated most efficiently in forward mode, i.e. by the tangent linear code of $H$. By contrast a product of the form $\mathbf{H}^T v$ is the (transpose of the) derivative of the scalar valued function $v^T H(x)$, which is evaluated most efficiently in reverse mode, i.e. by the adjoint of $H$. The scalar forward and reverse modes required for efficient evaluation of the above Jacobian-vector products are the standard forms of derivative code. The scalar mode is contrasted by the vector mode. In forward mode the vector mode simultaneously computes the sensitivities with respect to multiple input quantities, and in reverse mode simultaneously the sensitivity of multiple output quantities. Experience shows that the vector mode is considerably more efficient than multiple runs in scalar mode (see, e.g., Kaminski et al., 2003). We use the vector mode for applications that require the entire Jacobian, $\mathbf{H}$. Here the sensible choice between forward and reverse modes depends on the relative dimensions of state and observation spaces.

A particular advantage of AD is that it can guarantee readability and locality (Talagrand, 1991), i.e. every statement in the derivative code belongs to a particular statement in the function code. As a consequence, if the function code is modular, the same modularity is preserved in the derivative code. Another important advantage of the AD approach is that it simplifies the maintenance of the derivative code, because is can be quickly updated after any modification of the function code.

Since an AD tool operates at the code level, it is restricted to a particular programming language. For the most frequently used programming languages in Earth System Science, namely Fortran and C, AD tools are, however, available. It is a considerable effort to develop and maintain an AD tool up to a level robust enough for relevant scientific applications. Over the last decade, tool development has made good progress and there is a long list of successful AD applications to component models of the Earth System. A prime example is the above mentioned MITgcm, which is compliant to multiple AD tools (Forget et al., 2015). Typically an initial effort is required to achieve compliance of a model with an AD tool. From this basis, keeping this compliance for incremental updates of the model or the AD tool is less demanding. This is on one hand because AD tool developers, before each new release, apply regression tests against a set of benchmarking codes to preserve this compliance and, on the other hand, the incremental model updates typically respect the AD tool's coding requirements. Examples are limitations in the handling of pointers or memory allocation/deallocation sequences. It should also be noted that some AD tools allow the insertion of directives that support the analysis of the model code. These directives are helpful (and sometimes necessary) to enhance the efficiency of the generated code.

In some cases analytical formulations of the derivative can be derived and implemented with the observation operator (Moncet et al., 2008). Alternatively, the AD process can be mimicked by hand (see, e.g., Rabier et al., 2000; Weaver et al., 2003; Moore et al., 2004; Kleespies et al., 2004; O'Dell et al., 2006; Barrett and Renzullo, 2009), i.e. a human transforms the function code line by line into derivative code following the same recipes (Giering and Kaminski, 1998) that are implemented in AD tools. The advantage of hand-coding derivatives is that a human can be more flexible than a software tool. On the other hand the hand-coding approach is tedious and error prone. As a consequence this approach requires considerable development and maintenance effort, and is restricted to first derivatives. The large assimilation systems in the above list (Rabier et al., 2000; Weaver et al., 2003; Moore et al., 2004) were set up before AD tools were mature enough to handle the respective function codes.

Whether coded by hand or by an AD tool, the differentiation process typically reveals issues in the function code that are not apparent otherwise. A standard example is the square root, that is used, e.g., in the computation of the norm, the derivative of which tends to infinity as the argument tends to 0. Infinite sensitivities were typically not intended when the model code was designed and we can regard a differentiable reformulation as model improvement. In this context, it is helpful that AD tools (see, for example Pascual and Hascoët, 2008, for C and Fortran) support provision of derivative code for external routines. An example is the introduction of a floor value of 0 to avoid negative values of the simulated ice covered area. An obvious implementation as the maximum of the simulated area and 0 produces a step in the derivative at 0. Another example (now for the implementation as a minimum) is the formulation of co-limitation in biogeochemical models, in particular for carbon fixation in the photosynthesis model of Farquhar et al. (1980). Kaminski et al.

(2013) and Schürmann et al. (2016) describe the replacement of non-differentiabilities by smooth alternatives (including a look up table) in a model of the terrestrial biosphere. The handling of similar non-differentiabilities in a crop model is described by Lauvernet et al. (2012) .

Xu (1996) discusses treatment non-differentiabilities in NWP models through convection, precipitation or clouds, and Lorenc and Payne (2007) highlight the use of a statistical variational concept of 4-DVar at convective scale. Blessing et al. (2014) explore smoothing of non differentiabilities, in particular of the atmospheric component of an Earth System model.

Typical formulations of leaf phenology rely on a number of on/off switches that yield a non-differentiable behaviour and hamper the performance in a CCDAS. This problem was addressed by Knorr et al. (2010) through the design of an alternative phenology model that is based on a probabilistic approach and takes spatial sub-grid variability into account. As a consequence the model yields (more realistic) smooth transitions instead of on/off behaviour and performs well in CCDAS applications (Knorr et al., 2010; Kaminski et al., 2012a; Schürmann et al., 2016). Our recommendation is to consider the availability of the sensitivity information as an additional perspective on the model and its implementation. One can benefit from this sensitivity information and improve the modelling concept, as demonstrated for leaf phenology by Knorr et al. (2010) or remove errors in the implementation as described by Kaminski et al. (2003).

## 5 Conclusions

EO products can only be accessed by Earth system models via suitable observation operators. Hence careful design of observation operators is essential to optimally exploit the observational information. There are overlaps between observation operators used to confront dynamical models with EO data (validation, benchmarking, assimilation) and forward models used for retrievals of geophysical products. To allow a most flexible use, observation operators should be designed in modular form with carefully constructed interfaces. Several advanced retrieval algorithms and advanced assimilation techniques (Kalman Filter, 3D-Var, and 4D-Var) rely on first derivatives (linearisations) of the observation operators, i.e. their tangent and adjoint versions. Assessment of uncertainties and quantitative network design in addition require second derivatives of observation operators. To maximise their application range, these derivative codes should be developed and maintained together with their underlying observation operators. This procedure is, for example, applied at the European Centre for Medium range Weather Forecasting. Automatic Differentiation (AD) provides a means to minimise the development and maintenance effort for these derivative codes. There is an ever-increasing list of successful AD applications to large-scale Earth sciences codes, including many observation operators. Meanwhile there is a tendency among code developers to achieve and preserve compliance with an AD tool and thus enhance the functionality of their modelling system through the availability of derivative information. In the development of an AD-compliant modelling

or retrieval system, the system's sustainability can be maximised by the selection of a mature AD-tool that is permanently maintained by an experienced development team and extended in response to the evolution of user needs and programming languages. Close collaboration with AD-tool developers has proven beneficial in the efficient setup of robust AD-compliant systems for modelling (see, e.g., Rayner et al., 2005; Forget et al., 2015; Schürmann et al., 2016; Kaminski et al., 2016b) or retrieval (see, e.g., Pinty et al., 2007; Lauvernet et al., 2008, 2012; Lewis et al., 2012).

*Acknowledgements.* The authors would like to thank Laurent Bertino, Frédéric Chevallier, Patrick Heimbach, Christian Melsheimer, and Bernard Pinty, and three anonymous reviewers for helpful comments.. We acknowledge the support from the International Space Science Institute (ISSI). This publication is an outcome of the ISSI's Working Group on "Carbon Cycle Data Assimilation: How to consistently assimilate multiple data streams". T. Kaminski was in part funded by the ESA GHG-CCI project.

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
