# Peer review of "Flying the satellite into your model: on the role of observation operators in constraining models of the earth system and the carbon cycle"

_Biogeosciences, 2016_

## Referee Comment (RC1) · Anonymous Referee #1 · 29 Jun 2016

Review of Kaminski & Mathieu, «Reviews and Syntheses: Flying the Satellite into Your Model»

This review is potentially a useful addition to the literature. However, I think the authors need to do some work to make the paper a valuable contribution within the standards of what a review paper should be. In particular, the authors need to address the following:

   (i)     Incompleteness in presentation;
   (ii)    Statements not reflecting accurately current developments in the field of data assimilation;
   (iii)   Avoid slightly parochial references to data assimilation, e.g., the authors appear to focus too much on work done in the carbon cycle, with little or no reference to other areas of the Earth System;
   (iv)    The overall message from the paper should be stronger.

I provide examples of (i)-(iv) in the specific comments below, which the authors must address as well.

Specific comments:

L. 30: I think you should mention both spatial and temporal scales.

L. 150: I think the authors should specify that the observation operator $H$ is generally non-linear.

L. 154: The penalty function (or misfit function) in Eq. (2) can also have extra terms, e.g., constraints due to dynamics. Equation (3) represents the strong-constraint version of 4D-Var (this should be mentioned here). The authors introduce the weak-constraint formulation in L. 192.

L. 194: In principle, one could also have a linear operator **H** in Var.

L. 197: Identify the typical size of the state vector, e.g., for NWP this is of order $10^7$ elements.

L. 204: I suggest you provide other examples besides carbon dioxide.

L. 217: Particle filters have problems of filter degeneracy with high-dimensional problems, which makes difficult their application. Snyder et al. (2008) discusses this.

L. 240-283 (Section 3.2): The examples of data assimilation should reflect more areas than those mentioned by the authors. For example, they should include chemical data assimilation (for the stratosphere and the troposphere).

L. 249: Currently, efforts at the weather centres focus on hybrid approaches with combination of ensemble and variational methods (so it is not just 4D-Var as the text suggests). Such a hybrid approach has been operational at ECMWF for a while (Buizza et al., 2008; Isaksen et al., 2010; Bonavita et al., 2012), and is now operational, e.g., at the Met Office, UK, for the global model (Clayton et al., 2013) and at Environment Canada (Buehner et al., 2010). Discussion of these methods also took place at the 6th WMO Data Assimilation Symposium (http://das6.cscamm. umd.edu/). This is an example where to my mind, the text does not reflect recent developments in data asimilation, and the authors should modify the text.

L. 335: The authors focus is on the specification of the observational error covariance matrix. However, the specification of the background error covariance is a main difficulty in application of data assimilation to realistic problems. There are several reviews on this, including Bannister (2008a, b).

L. 360: The information provided by the text is incomplete. The authors should differentiate between OSEs (observing system experiments) and OSSEs (observing system simulation experiments). The authors should include reviews/overviews of OSSEs from the peer-reviewed literature. Two examples are Masutani et al. (2010), for general OSSEs, and Timmermans et al. (2015) for OSSEs for air quality observations. The authors should also provide more details on the shortcomings of OSSEs in L. 374 (or later in L. 392). The above reviews discuss these shortcomings.

L. 500-515: I suggest the authors provide more balance in their data assimilation examples. As the text reads to me, it shows a strong bias toward the carbon cycle.

L. 536-538: The final sentence in the paper seems weak. Overall, I do not see a strong message from the authors. They should address this. Also, who are these "experienced development teams"? Should they be part of the data assimilation setup at the weather centres? Elsewhere?

Typos, editorial:

L. 8: commonalities.

L. 12: One can misinterpret "derivative code" as code that is not original. I suggest something like "codes for differentiation".

L. 142: Do you need the word "advanced"?

L. 245: Check that you introduce acronyms in the paper.

L. 412 (and elsewhere): transposed -> transpose.

L. 463: Do you need the word "distinct"?

L. 466: Avoid the use of subjective statements like "luckily".

L. 498-499: I do not understand the phrase "A straight-forward…at 0". Perhaps reword.

References:

Bannister, R.N. (2008a). A review of forecast error covariance statistics in atmospheric variational data assimilation. I: characteristics and measurements of forecast error covariances. *Q. J. R. Meteorol. Soc*. 134, 1951–1970. doi: 10.1002/qj.339

Bannister, R.N. (2008b). A review of forecast error covariance statistics in atmospheric variational data assimilation. II: modelling the forecast error covariances. *Q. J. R. Meteorol. Soc.* 134, 1971–1996. doi: 10.1002/qj.340

Bonavita, M., Isaksen, L., and Hólm, E. (2012). "On the use of EDA background error variances in the ECMWF 4D-Var," in *ECMWF Tech Memo 664*. Available online at:http://www.ecmwf.int.

Buehner, M., Houtekamer, P.L., Charette, C., Mitchell, H.L., and He, B. (2010). Intercomparison of variational data assimilation and the ensemble kalman filter for global deterministic NWP. PartI: description and single-observation experiments. *Mon. Weather Rev.* 138, 1550–1566. doi: 10.1175/2009 MWR3157.1

Buizza, R., Leutbecher, M., and Isaksen,L. (2008). Potential use of an ensemble of analyses in the ECMWF Ensemble Prediction System. *Q. J. R. Meteorol. Soc.* 134, 2051–2066. doi: 10.1002/qj.346

Clayton, A.M., Lorenc, A.C., and Barker, D.M. (2013). Operational implementation of a hybrid ensemble/4D-Var global data assimilation system at the Met Office. *Q. J. R. Meteorol. Soc.* 139, 1445–1461. doi: 10.1002/qj.2054

Isaksen, L., Bonavita, M., Buizza, R., Fisher, M., Haseler, J., Leutbecher, M., et al. (2010). "Ensemble of data assimilations at ECMWF," in *ECMWF Tech Memo 636*. Available online at: http://www.ecmwf.int

Masutani, M., J.S. Woollen, S.J. Lord, G.D. Emmitt, T.J. Kleespies, S.A. Wood, S. Greco, H. Sun, J. Terry, V. Kapoor, R. Treadon and K.A. Campana (2010). Observing system simulation experiments at the national centers for environmental prediction. *J. Geophys. Res.*, 115, doi:10.1029/2009JD012528.

Snyder, C., Bengtsson, T., Bickel, P., and Anderson, J. (2008). Obstacles to high dimensional particle filtering. *Mon. Weather Rev.* 136, 4629–4640. doi: 10.1175/2008MWR2529.1

Timmermans, R., W.A. Lahoz, J.-L. Attié, V.-H. Peuch, L. Curier, D. Edwards, H. Eskes, and P. Builtjes (2015). Observing System Simulation Experiments for Air Quality. *Atmos. Env.*, 115, 199-213, doi:10.1016/j.atmosenv.2015.05.032

---

## Referee Comment (RC2) · Anonymous Referee #2 · 6 Sep 2016

\* General comments:

- This manuscript is largely technically correct and appropriate for Biogeosciences. The following comments are submitted for the consideration of the Authors; they aim at improving the readability and impact of this paper.

- The Authors may want to clarify the intended audience, and then to fine-tune their manuscript to provide added value for that particular audience. Indeed, it is likely that those readers who are fully familiar with model inversion and data assimilation will clearly understand the current version of the paper, but may not learn much from such a generic presentation. On the other hand, someone who has no background whatsoever in the subject matter may find it difficult to benefit from the manuscript, due to the idiosyncrasies noted below.

[Figure]

- It may also be helpful to revise the paper with a view to homogenize its various parts, and better link the technical information that is provided to an overall (partly missing) context. For instance, Sections 1 and 2 discuss 'observation operators', state variables and other concepts, but do not say much or anything at all about assimilation and retrieval (other than the rather enigmatic statement on Line 26). Section 3 then jumps into these latter methods, but does not explain why they are needed in the first place. Since models have been adjusted to data sets for centuries, way before "assimilation" was in vogue, there is a logical or thematic gap here that only specialists knowledgeable with the field will be able to leap over...

- In the same vein, the manuscript would gain from a more consistent use of mathematical symbols. It is counterproductive, in such a general paper, presumably aimed at a large audience, to keep switching notations or to assign different meanings to a particular symbol along the way. The paper also makes extensive use of acronyms, in particular to designate space instruments, but only a few are explicitly expanded. A simple approach to address this issue may be to provide the URL to an appropriate web site where further information can be found.

* Specific remarks:

- The title of the paper is somewhat dubious: It is not the satellite that flies into the model, but a satellite model (actually an observation operator) that is merged into another model. The drive to get a "catchy" title is understandable, but this one may not be particularly successful, or representative of what is actually discussed. Of course, this is a purely personal impression and a minor issue...

- Line 27: Most models require not only the specification of initial conditions but also boundary conditions, where empirical evidence plays a crucial role too.

- Lines 29-43: This text fragment is very clear about the difficulty of exploiting empirical data into a model that works on different space and time scales and resolutions. Yet, none of the subsequent discussion appear to refer back explicitly to these important

statements, for instance to explain how in practice the observation operator actually bridges the gaps between the instrument specifications, the observational protocol, and the modeling constraints. Again, depending on the expected audience, it may be pertinent to provide some more concrete information about the practicality of implementing such an approach.

- Lines 55-85: Section 2.1 is supposed to provide definitions of key concepts, but turns out to be rather obscure. For instance, the concepts of "state", "state variable", "state vector", "state space" are used without any explanation or context. Similarly, Section 3 discusses "model state", "observed state", etc. It may be useful to specify whether these expressions apply to the actual system under observation, or to the computer-simulated model. In any case, if the jargon of thermodynamics and systems dynamics is to be taken for granted, much of the rest of the paper might be of limited interest to those specialists. On the other hand, a clear introduction to these essential ideas could prove beneficial to readers unfamiliar with these concepts.

- Lines 65-67: The previous point is further highlighted by the unfortunate confusion between the state variables and prognostic variables. The former uniquely define the current "state" of the system, whether it evolves or not, and whether there is an attempt to predict its evolution or not. Prognostic variables are those that are forecast by a time-dependent model. These sets may be, but do not need to be identical. Clarity of mind is all the more important in this case because the general context of the discussion relates to the time evolution of complex systems such as the climate, or the carbon cycle, while the measurements obtained from satellites and assimilated through observation operators largely interpreted as repeated, but instantaneous snapshots (i.e., static processes), given the speed of propagation of electromagnetic waves compared to the rate of evolution of the system of interest.

- Lines 69-70: Similarly, the phrase "we must arrange for a change of the state" could be potentially ambiguous, as the actual state of the system is (usually) not change-able: instead, the "state variables" that describe the state of the simulation model are

modified to reduce the "distance" between the simulated state (of the model) and the measurements, which in principle describe the state of the actual (observed) system.

- Figure 2: The right side of the graph is truncated: either move or re-design the whole graphics.

- Subsections 2.2, 3.2 and 3.3 enumerate multiple examples or applications that make use of observation operators, data assimilation techniques and retrieval methods. Such lists convey the message that these techniques are indeed exploited in a variety of fields, but do not really contribute to a better understanding of the design, development or functioning of such a tool. The point is not to delete these sections, but to clarify their purposes and added value: if they are meant to be a review of the field, then they may need to be beefed up. But if the intent is only to indicate that these techniques are widely used in a range of disciplines, then shorter sections or pointers to the literature may suffice.

- Line 121: Please note that both older (2004) and more recents (2013, 2015) publications have appeared on RAMI: see http://rami-benchmark.jrc.ec.europa.eu/HTML/RAMI-IV/RAMI-IV.php

- Line 154: If the purpose of the paper is to popularize advanced concepts as hypothesized above, then it might be appropriate to remark that the first term of Equation 2 is basically an expression of the least squares fit, which would be familiar to a broad range of readers. Similarly, the actual role and purpose of the second term of Equations (2) and (3) should be explained in more detail: Why is the first term insufficient? Could one rely on the second term only? Prospective users would likely gain from an understanding of these overall strategic questions before adopting these "advanced" methods.

- Lines 155-156: This enigmatic discussion about using x instead of z may be either pedantic or confusing: the symbols used in equations are irrelevant, as long as they are used systematically and coherently. Changing conventions in the middle of the paper

is unjustified, especially given the minimalist use of the symbol z anyway. Please use the simplest set of mathematical symbols throughout the paper.

- Line 369: Delete the extra "around" near the end of the line.

- Line 400: What exactly is the implication of the phrase "because the dimension of the control space is large"? Is there a choice to minimize with respect to any other variable? Or would a different variable be chosen if the dimension of the control space were smaller, whatever that means?

- Line 405: It is not immediately apparent why Equation (9) is the gradient of Equation (2). It may be useful to be somewhat more explicit, or to point the reader to a more detailed (preferably publicly available) source.

- Lines 421-423: This statement about the limited accuracy of "the above listed algorithms" may need a bit more substantiation: indeed, the more advanced methods described here also have a limited accuracy, so the issue revolves around demonstrating that the uncertainty associated with AD is always lower than that of other approaches. While it is true that "incorrect gradient information will slow down or prematurely stop the iterative minimisation of J", why would those other methods systematically yield incorrect, or "less correct" gradients?

- Lines 423-424: Similarly, the statement concerning "the computational cost of this approximation grows linearly with the length of the control vector" may be true, but needs to be evaluated against a similar statement about what controls the cost of the AD method. Again, these claims may be correct, but they should be substantiated.

- Line 499: The phrase "maximum of the simulated area and 0 produces are step in the derivative at 0" is confusing or ill-stated.

- Line 502: Similarly, the phrase "describe to replacement" is odd.

- Line 520: The claim that "EO products can only be accessed by Earth system models via suitable observation operators" may be exaggerated: clearly, many users of EO

products carry on without relying on, or even knowing about, observation operators. What may be more appropriate is to discuss why and to what extent such operators and the associated methods of assimilation/inversion provide more satisfactory results than traditional or earlier methods. An effective way to achieve this goal is to demonstrate the drawbacks that may arise when exploiting remote sensing data without relying on such advanced techniques.

I hope these comments may be helpful in updating the manuscript.

---

## Author Comment (AC1) · 26 Dec 2016

[10pt]article [authoryear,round]natbib [normalem]ulem color

We thank the reviewers for their careful inspection of the manuscript. Their comments were very helpful in improving the clarity of the manuscript.

In the following we address their comments point-by-point. We use *text in italics* to repeat the reviewer comments, default font for our response, and **bold faced text** for quotations from the manuscript, with changes marked in colour.

We provide the revised manuscript (with and without changes highlighed) in the supplement.

[Figure]

**1 comments by Anonymous Referee #1**

1. *This review is potentially a useful addition to the literature. However, I think the authors need to do some work to make the paper a valuable contribution within the standards of what a review paper should be. In particular, the authors need to address the following: (i) Incompleteness in presentation; (ii) Statements not reflecting accurately current developments in the field of data assimilation; (iii) Avoid slightly parochial references to data assimilation, e.g., the authors appear to focus too much on work done in the carbon cycle, with little or no reference to other areas of the Earth System; (iv) The overall message from the paper should be stronger. I provide examples of (i)-(iv) in the specific comments below, which the authors must address as well.*

   We'll address this comment by responding below specifically to the examples. In summary we clarify that our focus is on applications to the carbon cycle (which addresses the revier's points (i)-(iii)), we refer to another contribution to this special issue that addresses QND (which addresses points (i) and (ii)), we have followed several suggestions to include extra material/references (which addresses points (i)-(iii)), and we have clarified the message in the last sentences of the conclusions (which addresses point (iv)).

2. *L. 30: I think you should mention both spatial and temporal scales.*

   We think this, too, and explicitly had mentioned both in the sentence directly following the one on L 30. To make this even clearer we now also added 'space and time' in brackets to the sentence on L 30:

   **Such confrontation with observations is hampered by the fact that observed and modelled quantities typically differ in nature or scale (in space and time). For example, a flask sample of the atmospheric carbon dioxide concentration provides a value at a specific point in space and time,**

whereas an atmospheric tracer model operates in a discretised representation of space and time, i.e. on values that refer to a box in the atmosphere and a particular period of time.

3. *L. 150: I think the authors should specify that the observation operator H is generally non-linear.*

Good point, we found a good place for this further above:

**The link from the model to the observations is provided through a set of relationships expressed in terms of an *observation operator*. We can think of an observation operator as an arm, which enables the ESM to access a particular type of observation (see Figure 1). Observation We stress that the usage of the term *operator* is not meant to imply the linearity of the observation operator. In fact observation operators are ranging in complexity from a simple interpolation or integral scheme up to a chain of sophisticated non-linear radiative transfer models.**

4. *L. 154: The penalty function (or misfit function) in Eq. (2) can also have extra terms, e.g., constraints due to dynamics. Equation (3) represents the strong-constraint version of 4D-Var (this should be mentioned here). The authors introduce the weak-constraint formulation in L. 192.*

Regarding potential extra terms in the misfit function that include more sophisticated forms of prior information one may wish to bring into the assimilation procedure (such as dynamical constraints) we prefer not to add this extra level of complexity, in order not to confuse the reader.

For the same reason we prefer to start with the strong constraint variational approach without mentioning the weak constraint form (yet!).

5. *L. 194: In principle, one could also have a linear operator H in Var.*

Yes, as a special case.

6. *L. 197: Identify the typical size of the state vector, e.g., for NWP this is of order $10^7$ elements.*

Done:

**This allows an analytic solution $x_{po}$ of equation 2:**

$$x_{po} = x_{pr} - U_{xpo}H^T U_y^{-1}\left(Hx_{pr} - y_i\right) \qquad (5)$$

**the evaluation of which involves the inversion of the (typically high dimensional) matrix**

$$U_{xpo} = (H^T U_y^{-1} H + U_{xpr}^{-1})^{-1} \qquad (6)$$

**which is typically of high dimension (e.g., $10^7$ in NWP) and expresses the uncertainty range in $x_{po}$ that is consistent with uncertainty ranges in the data and the prior values.**

7. *L. 204: I suggest you provide other examples besides carbon dioxide.*

This special issue is dedicated to "Data assimilation in carbon/biogeochemical cycles: consistent assimilation of multiple data streams". This is why we put a special focus an carbon dioxide, but we realise that we did not make this sufficiently clear in the manuscript. Now we mention this at several places, including the abstract:

**Throughout this paper, as special focus lies on applications to the carbon cycle.**

8. *L. 217: Particle filters have problems of filter degeneracy with high-dimensional problems, which makes difficult their application. Snyder et al. (2008) discusses this.*

That's an important point, thanks:

**The feasible ensemble size is limited by the computational demands which are essentially determined by the complexity of the underlying model. We also note the challange of filter degeneracy that limits the applicability of particle filters to high-dimensional problems (see, e.g., Snyder et al., 2008) .**

9.  *L. 240-283 (Section 3.2): The examples of data assimilation should reflect more areas than those mentioned by the authors. For example, they should include chemical data assimilation (for the stratosphere and the troposphere).*

    See comment to point 7 above on the carbon cycle focus.

10. *L. 249: Currently, efforts at the weather centres focus on hybrid approaches with combination of ensemble and variational methods (so it is not just 4D-Var as the text suggests). Such a hybrid approach has been operational at ECMWF for a while (Buizza et al., 2008; Isaksen et al., 2010; Bonavita et al., 2012), and is now operational, e.g., at the Met Office, UK, for the global model (Clayton et al., 2013) and at Environment Canada (Buehner et al., 2010). Discussion of these methods also took place at the 6th WMO Data Assimilation Symposium (http://das6.cscamm. umd.edu/). This is an example where to my mind, the text does not reflect recent developments in data asimilation, and the authors should modify the text.*

    Thanks, this is a very good point, and we included it:

    **A recent development at NWP centres are hybrid approaches that combine ensemble and variational approaches. Such a hybrid approach is operational, e.g., at ECMWF (Buizza et al., 2008; Isaksen et al., 2010; Bonavita et al., 2012) or the NWP centres of the UK (Clayton et al., 2013) or Canada (Buehner et al., 2010).**

11. *L. 335: The authors focus is on the specification of the observational error covariance matrix. However, the specification of the background error covariance is a*

*main difficulty in application of data assimilation to realistic problems. There are several reviews on this, including Bannister (2008a, b).*

Our focus is on assimilation of multiple data streams which directly relates to the specification of the observational uncertainty. We included this point now, but not in this retrieval section but where we mention the cyclic setup in NWP:

**Operationally the assimilation scheme is run in cyclic mode through these two steps. In such a cyclic scheme, the prior information is provided by the previous forecast, i.e. it is consistent with the dynamical information from the model, and at the same time suffers from errors in the model. In this setup, the specification of the prior uncertainty is particularly challenging (see, e.g., Bannister, 2008a, b).**

12. *L. 360: The information provided by the text is incomplete. The authors should differentiate between OSEs (observing system experiments) and OSSEs (observing system simulation experiments). The authors should include reviews/overviews of OSSEs from the peer-reviewed literature. Two examples are Masutani et al. (2010), for general OSSEs, and Timmermans et al. (2015) for OSSEs for air quality observations. The authors should also provide more details on the shortcomings of OSSEs in L. 374 (or later in L. 392). The above reviews discuss these shortcomings.*

We refer now explicitly to a separate contribution to this special issue that is dedicated to QND, and we have also added a sentence of OSEs and the two suggested references:

**Observing System Simulation Experiments (OSSEs) and Quantitative Network Design (QND) are two methodologies that evaluate observation impact on assimilation systems. By an observing system or observational network we understand the superset of all observations that are made available to an assimilation system. We only give a brief introduction to the**

**topic, as QND for the carbon cycle is addressed by another contribution to this special issue.**

**An OSSE**  (see, e.g., Böttger et al., 2004; Masutani et al., 2010; Timmermans et al., 2015) **uses a model plus observation operators to simulate in a model analogues of observations that would be collected by a potential observing system (often the current observing system extended by a potential new data stream). The model is also used to simulate, in a so-called "nature run", a surrogate of reality, i.e. a reference trajectory over the period of investigation. Then an assimilation/forecast system (preferably built around** around **a different model) is used to evaluate some measure of the performance of the potential observing system and its sub-systems. In NWP, the performance of an observing system is usually quantified by the quality (skill) of a forecast from the initial value that was constrained by the observation system. Via this procedure one can, for example, assess the added value of a planned mission in terms of an increment in forecast skill. We also note a related approach, Observing System Experiments (OSEs), which assess observation impact by removing one or several *existing* data streams from the list of observations used in a data assimilation system.**

13. *L. 500-515: I suggest the authors provide more balance in their data assimilation examples. As the text reads to me, it shows a strong bias toward the carbon cycle.*

    See comment to point 7 above on the carbon cycle focus.

14. *L. 536-538: The final sentence in the paper seems weak. Overall, I do not see a strong message from the authors. They should address this. Also, who are these "experienced development teams"? Should they be part of the data assimilation setup at the weather centres? Elsewhere?*

We clarified the message of the final sentences:

**Meanwhile there is a tendency among code developers to achieve and preserve compliance with an automatic differentiation AD tool and thus enhance the functionality of their modelling system through the availability of derivative information.** To maximise sustainability of such a modelling system, it is essential that the automatic differentiation tool In the development of an AD-compliant modelling or retrieval system, the system's sustainability can be maximised by the selection of a mature AD-tool that **is permanently maintained** and adapted to user needs **by an experienced development team** . and extended in response to the evolution of user needs and programming languages. Close collaboration with AD-tool developers has proven beneficial in the efficient setup of robust AD-compliant systems for modelling (see, e.g., Rayner et al., 2005; Forget et al., 2015; Schürmann et al., 2016; Kaminski et al., 2016) or retrieval (see, e.g., Pinty et al., 2007; Lauvernet et al., 2008, 2012; Lewis et al., 2012) .

15. *Typos, editorial: L. 8: commonalities.*

    *L. 12: One can misinterpret "derivative code" as code that is not original. I suggest something like "codes for differentiation".*

    Thanks. Both fixed.

16. *L. 142: Do you need the word "advanced"?*

    We think yes. Given that there are extremely heuristic retrieval schemes and data assimilation schemes as simple as nudging, which have not so much in common.

17. *L. 245: Check that you introduce acronyms in the paper.*

    *L. 412 (and elsewhere): transposed -> transpose.*

    *L. 463: Do you need the word "distinct"?*

*L. 466: Avoid the use of subjective statements like "luckily".*

*L. 498-499: I do not understand the phrase "A straight-forward...at 0". Perhaps reword.* All fixed.

**2 comments by Anonymous Referee #2**

1. *\* General comments: - This manuscript is largely technically correct and appropriate for Biogeosciences. The following comments are submitted for the consideration of the Authors; they aim at improving the readability and impact of this paper. - The Authors may want to clarify the intended audience, and then to fine-tune their manuscript to provide added value for that particular audience. Indeed, it is likely that those readers who are fully familiar with model inversion and data assimilation will clearly understand the current version of the paper, but may not learn much from such a generic presentation. On the other hand, someone who has no background whatsoever in the subject matter may find it difficult to benefit from the manuscript, due to the idiosyncrasies noted below. - It may also be helpful to revise the paper with a view to homogenize its various parts, and better link the technical information that is provided to an overall (partly missing) context. For instance, Sections 1 and 2 discuss 'observation operators', state variables and other concepts, but do not say much or anything at all about assimilation and retrieval (other than the rather enigmatic statement on Line 26). Section 3 then jumps into these latter methods, but does not explain why they are needed in the first place. Since models have been adjusted to data sets for centuries, way before "assimilation" was in vogue, there is a logical or thematic gap here that only specialists knowledgeable with the field will be able to leap over... - In the same vein, the manuscript would gain from a more consistent use of mathematical symbols. It is counterproductive, in such a general paper, presumably aimed at a large audience, to keep switching notations or to assign different meanings*

*to a particular symbol along the way. The paper also makes extensive use of acronyms, in particular to designate space instruments, but only a few are explicitly expanded. A simple approach to address this issue may be to provide the URL to an appropriate web site where further information can be found.*

The target audience are scientists with some background in modelling and/or remote sensing. We have extended the manuscript at several places to improve its readability (see also response to detailed comments below). We also address the reviewer's point on consistent use of mathematical symbols in our response to the specific point 5 below. Further we have included explanations for acronyms and have removed some the acronyms to space instruments, where they were not needed.

2. *The title of the paper is somewhat dubious: It is not the satellite that flies into the model, but a satellite model (actually an observation operator) that is merged into another model. The drive to get a "catchy" title is understandable, but this one may not be particularly successful, or representative of what is actually discussed. Of course, this is a purely personal impression and a minor issue...*

We understand the reviewer's concern, but are both very fond of this title and would like to keep it.

3. *Line 27: Most models require not only the specification of initial conditions but also boundary conditions, where empirical evidence plays a crucial role too.*

Yes, this is correct in general. And we mention boundary values explicitly where we introduce the control vector (see point 12 below). In the context of an ESM boundary values may be less important than for component models. So, at this place, we left it with the two examples, initial conditions and process parameters, (hence we use "e.g."):

**A further step towards the rigorous use of the observations is their ingestion in formal data assimilation procedures, e.g. to constrain the model's**

**initial state (initialisation) or tunable parameters in the model's process representations (calibration).**

4. *Lines 29-43: This text fragment is very clear about the difficulty of exploiting empirical data into a model that works on different space and time scales and resolutions. Yet, none of the subsequent discussion appear to refer back explicitly to these important statements, for instance to explain how in practice the observation operator actually bridges the gaps between the instrument specifications, the observational protocol, and the modeling constraints. Again, depending on the expected audience, it may be pertinent to provide some more concrete information about the practicality of implementing such an approach.*

Thanks, we have taken up this point where we discuss $U_y$:

**The function $J(x)$ is composed of two terms. The first term quantifies the misfit between the observations and their simulated counterpart (observational term). $U_y$ has to account for the uncertainty in the observations and the uncertainty imposed by imperfection of the model, including the above-mentioned representativeness in space and time. For diagonal $U_y$ (uncorrelated uncertainty) it reduces to a least squares fit to the observations.**

5. *Lines 55-85: Section 2.1 is supposed to provide definitions of key concepts, but turns out to be rather obscure. For instance, the concepts of "state", "state variable", "state vector", "state space" are used without any explanation or context. Similarly, Section 3 discusses "model state", "observed state", etc. It may be useful to specify whether these expressions apply to the actual system under observation, or to the computersimulated model. In any case, if the jargon of thermodynamics and systems dynamics is to be taken for granted, much of the rest of the paper might be of limited interest to those specialists. On the other hand, a clear introduction to these essential ideas could prove beneficial to read-*

*ers unfamiliar with these concepts.*

This is a good point. We have made clear in the text now, that we deal with the state in the model.

**Mathematically the observation operator is defined as a mapping $H$ from the vector of state variables $z$ (of the model) onto the vector of observations $y$:**

$$H : z \mapsto y \tag{1}$$

**The observation vector can include, for example, observed radiances, radar backscatter, or in situ observations. The vector of the model's state variables (*state vector*) defines the simulated system for a given time step at all points in space, and the evolution of the system is described by a sequence of state vectors, forming a trajectory through the state space.**

And we also avoided the unfortunate expression "observed state" in section 3.

6. *Lines 65-67: The previous point is further highlighted by the unfortunate confusion between the state variables and prognostic variables. The former uniquely define the current "state" of the system, whether it evolves or not, and whether there is an attempt to predict its evolution or not. Prognostic variables are those that are forecast by a timedependent model. These sets may be, but do not need to be identical. Clarity of mind is all the more important in this case because the general context of the discussion relates to the time evolution of complex systems such as the climate, or the carbon cycle, while the measurements obtained from satellites and assimilated through observation operators largely interpreted as repeated, but instantaneous snapshots (i.e., static processes), given the speed of propagation of electromagnetic waves compared to the rate of evolution of the system of interest.*

See response to point 5 above. And we also made the following clarification:

**The state variables of a dynamical model are also called prognostic variables, to contrast them with diagnostic variables, which are computed from the state and evolve only indirectly through the evolution of the state. For example the albedo of the land surface is diagnosed from the state of the vegetation-soil system. Hence, if we want to change the trajectory of the model achieve a change in the model state at any given point in time, the model will then propagate this change of state forward in time, and we achieve a change of the model trajectory (e.g. to improve the fit to observations), we must arrange for a change of the state. The model will then propagate this change of state forward in time. . This means, to bring observational information into the model, we must link the observations to the state: In other words the model's state vector constitutes the interface between the model and the observation operator.**

7. *Lines 69-70: Similarly, the phrase "we must arrange for a change of the state" could be potentially ambiguous, as the actual state of the system is (usually) not changeable: instead, the "state variables" that describe the state of the simulation model are modified to reduce the "distance" between the simulated state (of the model) and the measurements, which in principle describe the state of the actual (observed) system.*

We clarified in the text, see point 7 above.

8. *Figure 2: The right side of the graph is truncated: either move or re-design the whole graphics.*

Thanks, is corrected.

9. *Subsections 2.2, 3.2 and 3.3 enumerate multiple examples or applications that make use of observation operators, data assimilation techniques and retrieval methods. Such lists convey the message that these techniques are indeed exploited in a variety of fields, but do not really contribute to a better understanding*

*of the design, development or functioning of such a tool. The point is not to delete these sections, but to clarify their purposes and added value: if they are meant to be a review of the field, then they may need to be beefed up. But if the intent is only to indicate that these techniques are widely used in a range of disciplines, then shorter sections or pointers to the literature may suffice.*

Our objective in section 2.2. is to explain that observation operators are widespread. And our objective in sections 3.2. and 3.3 is to show the commonalities. We clarified this in the beginning of section 3:

**This section starts with an introduction of the formalism behind advanced data assimilation and retrieval schemes. The details of the formalism are useful to understand the application examples in this section and the (and the commonalities between assimilation and retrievals) and the need for derivative information that is discussed in section 4.**

10. *Line 121: Please note that both older (2004) and more recents (2013, 2015) publications have appeared on RAMI: see http://ramibenchmark. jrc.ec.europa.eu/HTML/RAMI-IV/RAMI-IV.php*

Thanks, references to 2013 and 2015 publications added.

11. *Line 154: If the purpose of the paper is to popularize advanced concepts as hypothesized above, then it might be appropriate to remark that the first term of Equation 2 is basically an expression of the least squares fit, which would be familiar to a broad range of readers. Similarly, the actual role and purpose of the second term of Equations (2) and (3) should be explained in more detail: Why is the first term insufficient? Could one rely on the second term only? Prospective users would likely gain from an understanding of these overall strategic questions before adopting these "advanced" methods.*

Both very good suggestions, thanks. Both included:

**$U_y$ has to account for the uncertainty in the observations and the un-certainty imposed by imperfection of the model, including the above-mentioned representativeness in space and time. For diagonal $U_y$ (uncorre-lated uncertainty) it reduces to a least squares fit to the observations.** The second term quantifies the deviation of the model state from the prior infor-mation $x_{pr}$ (prior term, often also called background).  **This term provides a means to include information in addition to the observational information into the assimilation procedure and it ensures the existence of a minimum in cases where the observational information is not sufficient to constrain the unknowns. Both terms, observation misfit and prior,** are weighted in inverse proportion to the respective uncertainties, i.e. the com-bined uncertainty in the observations and observation operator, $U_y$, and the uncertainty in the prior information, $U_{xpr}$. The superscript $T$ denotes trans-position. Note that the equation does not require the observations to be provided in the space time grid of the model.

12. *Lines 155-156: This enigmatic discussion about using x instead of z may be ei-ther pedantic or confusing: the symbols used in equations are irrelevant, as long as they are used systematically and coherently. Changing conventions in the middle of the paper is unjustified, especially given the minimalist use of the sym-bol z anyway. Please use the simplest set of mathematical symbols throughout the paper.*

In fact, it is exactly to achieve clarity in concept and notation why we need to distiguish between state vector and the vector of unknowns. We made this clear:

**The assimilation problem is typically formalised as a minimisation problem for a misfit function**

$$J(x) = \frac{1}{2} \left(H(x) - y\right)^T U_y^{-1} \left(H(x) - y\right) + \frac{1}{2} \left(x - x_{pr}\right)^T U_{xpr}^{-1} \left(x - x_{pr}\right) \quad , \quad (2)$$

where  $x$  denotes the vector of un- instead of reusing the above defined symbol for the . Even though in some applications this vector of unknowns may coincide with the model state, $z$. This is more convenient for later use where $x$ is more general than the state, this is not generally the case (as will be discussed below), and we need to make a clear distinction between both objects.

13. *Line 369: Delete the extra "around" near the end of the line.*

    Thanks, is corrected.

14. *Line 400: What exactly is the implication of the phrase "because the dimension of the control space is large"? Is there a choice to minimize with respect to any other variable? Or would a different variable be chosen if the dimension of the control space were smaller, whatever that means?*

    Rephrased:

    In variational assimilation, equation 2 or equation 3 are typically minimised in an iterative procedure that varies $x$, . To do this efficiently even for high-dimensional control spaces, so-called gradient algorithms are employed. They rely on the capability of evaluating the gradient of $J$ with respect to $x$ to define a search direction in the space of unknowns. The gradient is useful, because it yields the direction of steepest ascent. For $J(x)$ of equation 2  straight-forward differentiation with respect to $x$ yields (see, e.g., section 3.4.4 of Tarantola (2005)) the gradient

$$\nabla J(x) = H(x)^T U_y^{-1}\left(H(x) - y\right) + U_{xpr}^{-1}\left(x - x_{pr}\right),\qquad(9)$$

    and we see that its evaluation requires the capability to multiply the  transpose of $H$ with a vector.

15. *Line 405: It is not immediately apparent why Equation (9) is the gradient of Equation (2). It may be useful to be somewhat more explicit, or to point the reader to a more detailed (preferably publicly available) source.*

Added explanatory text (see point 14 above)

16. *Lines 421-423: This statement about the limited accuracy of "the above listed algorithms" may need a bit more substantiation: indeed, the more advanced methods described here also have a limited accuracy, so the issue revolves around demonstrating that the uncertainty associated with AD is always lower than that of other approaches. While it is true that "incorrect gradient information will slow down or prematurely stop the iterative minimisation of J", why would those other methods systematically yield incorrect, or "less correct" gradients?*

Added explanatory text:

**Traditionally derivatives were approximated by multiple forward runs (finite difference approximation) (see, e.g., Toudal, 1994; Melsheimer et al., 2009; Govaerts et al., 2010; Dubovik et al., 2011). This discretised procedure has two disadvantages: The first is the limited accuracy** of this gradient approximation (providing only the linear term of the Taylor series), **which degrades the performance of the above listed algorithms. For example, incorrect gradient information will slow down or prematurely stop the iterative minimisation of** $J$**,** because gradient-based minimisation algorithms rely on the consistency of evaluations of $J$ and its gradient**. The other disadvantage is that the computational cost of this approximation grows linearly with the length of the control vector.**

17. *Lines 423-424: Similarly, the statement concerning "the computational cost of this approximation grows linearly with the length of the control vector" may be true, but needs to be evaluated against a similar statement about what controls*

*the cost of the AD method. Again, these claims may be correct, but they should be substantiated.*

This substantiation happens in the middle of the following paragraph.

**... By contrast, the CPU time required by the adjoint code is proportional to the number of output variables and independent of the number of input variables. ...**

18. *Line 499: The phrase "maximum of the simulated area and 0 produces are step in the derivative at 0" is confusing or ill-stated.*

We are sorry, there was a typo in the middle of the sentence, and we also edited the text around a bit:

**An example is the introduction of a floor value of 0 to avoid negative values of the simulated ice covered area. A straight-forward An obvious implementation as the maximum of the simulated area and 0 produces are a step in the derivative at 0. Another example (now for the implementation as a minimum) is the formulation of co-limitation in biogeochemical models, in particular for carbon fixation in the photosynthesis model of Farquhar et al. (1980). Kaminski et al. (2013) and Schürmann et al. (2016) describe to the replacement of non-differentiabilities by smooth alternatives (including a look up table) in a model of the terrestrial biosphere.**

19. *Line 502: Similarly, the phrase "describe to replacement" is odd.*

We are sorry, here there was another typo, which we corrected (see point 18 above).

20. *Line 520: The claim that "EO products can only be accessed by Earth system models via suitable observation operators" may be exaggerated: clearly, many users of EO products carry on without relying on, or even knowing about, observation operators. What may be more appropriate is to discuss why and to*

*what extent such operators and the associated methods of assimilation/inversion provide more satisfactory results than traditional or earlier methods. An effective way to achieve this goal is to demonstrate the drawbacks that may arise when exploiting remote sensing data without relying on such advanced techniques.*

We agree that EO data can be exploited without knowing about the term observation operators, but still they apply observation operators, which in some cases can be as basic as regridding.

21. *I hope these comments may be helpful in updating the manuscript.*

These comments were very helpful, many thanks.

---

## Author Response (AR1)

Following the suggestion of the editor we have modified the title such that it explicitly mentions "observation operator" and the "carbon cycle", "earth system" is mentioned first, because most of the examples go beyond the carbon cycle. It now reads:

**Flying the satellite into your model: on the role of observation operators in constraining models of the earth system and the carbon cycle**

We have further added one cross-reference to another contribution to this special issue and corrected the format of two references.